# Genetic profiling of inherited colorectal cancer syndromes in Tunisian patients

Rania Abdelmaksoud-Dammak[1], Nihel Ammous-Boukhris[1], Amena Saadallah-Kallel[1], Dorra Ben Ayed-Guerfali[1], Souhir Guidara[2], Imen Miladi-Abdennadher[1], Ikhlas Ben Ayed[2], Hassen Kamoun[2], Slim Charfi[3], Tahya Sellami-Boudawara[3], Imen Zribi[4], Foued Frikha[4], Salah Boujelbene[4], Raja Mokdad-Gargouri[1]*

1 Center of Biotechnology of Sfax, Laboratory of Eukaryotes Molecular Biotechnology. University of Sfax, Sfax, Tunisia, 2 Department of Human Genetics, Hedi Chaker Hospital, University of Sfax, Sfax, Tunisia, 3 Department of Anatomo-pathology, Habib Bourguiba Hospital, University of Sfax, Sfax, Tunisia, 4 Department of Surgery, Habib Bourguiba Hospital, University of Sfax, Sfax, Tunisia

* raja.gargouri@cbs.rnrt.tn

## Abstract

### Objective

Colorectal cancer (CRC) is among the most commonly diagnosed cancers worldwide, with 2% to 5% of cases being linked to inherited syndromes.

### Material and methods

A cohort of 30 Tunisian patients was selected and divided into two groups based on clinical features and family history: Group 1 included patients clinically diagnosed with hereditary polyposis syndromes, including MUTYH-Associated Polyposis (MAP: 15 cases) and Familial Adenomatous Polyposis (FAP: 5 cases). Group 2 consisted of patients clinically diagnosed with non-polyposis syndromes, including Lynch Syndrome (LS: 7 cases) and other rare syndromes (OS: 3 cases). Genetic testing was performed using either Sanger sequencing or targeted next-generation sequencing (NGS) with a cancer panel including 31 cancer-related genes.

### Results

In Group 1, MAP was confirmed in 13 patients who were homozygous carriers of the pathogenic variant (c.1143_1144dup p.Glu382fs) in the *MUTYH* gene. For patients suspected of having FAP, pathogenic variants in the *APC* gene were identified in only two patients (c.3183_3187del p.Lys1061_Gln1062insTer, and c.2016_2017del p.His672Ter), while another patient carried a frameshift variant (c.502_503del, p.Ile-168SerTer11) in the *PTEN* gene, indicating Cowden Syndrome. In Group 2, genetic testing confirmed Peutz-Jeghers Syndrome in a young girl who had a large deletion in the *STK11* gene. For patients suspected to have LS, only variants of unknown

**Data availability statement:** The Fastq files generated in this study were deposited in the NCBI database under the Accession Number: "PRJNA1258780".

**Funding:** Tunisian Ministry of Higher education and Scientific Research. The funders had no role in study design, data collection and analysis, decision to publish, or preparation of the manuscript.

**Competing interests:** The authors have declared that no competing interests exist.

**Abreviations:** Polyp = Polyposis, BC = Breast Cancer, BlC = Bladder Cancer, BrC = Brain Cancer, CHRPE = Congenital hypertrophy of the retinal pigment epithelium, CRC = ColoRectal Cancer, EC = Endometrial Cancer, GC = Gastric Cancer, HC = Hepatic Cancer, HmC = Haematological Cancer, KC = Kidney Cancer, L =Leukemia, LC = Lung Cancer, LxC = Larynx Cancer, NPC = Nasopharyngeal Carcinoma, Os = Osteosarcoma, OvC = Ovarian Cancer, PC = Pancreatic Cancer, PpC = Papillary Cancer, PrC = Prostate Cancer, TC = Tongue Cancer, ThC = Thyroid Cancer, UC = Uterin Cancer, UrC = ureteral cancer, ND = Not Determined.

significance (VUS) were identified in MMR. Further genetic investigations are required to identify the pathogenic variant in these patients.

## Conclusion

Overall, our results highlight the importance of genetic testing to better understand hereditary CRC syndromes in Tunisian families, and to improve the management of patients and their relatives.

---

## Introduction

Colorectal cancer (CRC) is the third most commonly diagnosed cancer worldwide and the second leading cause of cancer-related deaths globally [1]. The implementation of CRC screening for adults aged 50 and over has led to a reduction in both the incidence and mortality associated with late-onset CRC [2,3]. About 10% of newly diagnosed CRC cases are identified in individuals under 50, highlighting it as an emerging health concern [4].

CRC results from a combination of genetic and environmental factors, making it a complex disease [5]. Several factors such as obesity, high-fat diets, and smoking, increase the risk of CRC [6]. Approximately 30% of all CRC cases are believed to have an inherited component, often due to a first-degree relative with this malignancy [7,8].

Inherited CRC syndromes are categorized into those with polyposis and those without polyposis. Familial Adenomatous Polyposis (FAP) is the most common polyposis syndrome, while Lynch Syndrome (LS) is the most common non-polyposis syndrome [9]. These highly penetrant CRC predisposition syndromes are associated with a high lifetime risk of developing CRC, ranging from 15–52% for LS to 100% for classic FAP [7–9].

LS is characterized by an autosomal dominant mode of inheritance and an increased risk of extracolonic cancers, including gynaecological (endometrial and ovarian), gastrointestinal, and genitourinary cancers, as well as sebaceous adenomas and skin carcinomas. Germline alterations include pathogenic variants in the mismatch repair (MMR) genes *MLH1*, *MSH2*, *MSH6*, and *PMS2*, or deletions in *EPCAM*, which lead to a loss of functional MMR proteins and result in microsatellite instability (MSI) [10]. LS-associated CRC tends to develop at younger ages (<50 years) and progress more rapidly compared with sporadic CRC [11].

FAP is the second most common inherited CRC syndrome, affecting approximately 1 in 10,000 individuals [7]. Germline pathogenic variants in the tumor suppressor gene *APC* follow an autosomal dominant inheritance pattern, leading to the development of FAP and the AFAP that is the attenuated form, [12]. Additionally, MUTYH-Associated Polyposis (MAP) is recognized as the second most common adenomatous polyposis syndrome. It is an autosomal recessive condition with high penetrance, linked to biallelic germline variants in the *MUTYH* gene [13]. Carriers with biallelic *MUTYH* variants have a significantly elevated lifetime risk of developing

CRC. However, it remains unclear whether individuals with a monoallelic pathogenic variant of *MUTYH* have an increased genetic susceptibility to CRC [14,15].

In addition to polyposis syndromes, three main types of gastrointestinal hereditary hamartomatous polyposis syndromes are noteworthy: Peutz–Jeghers Syndrome (PJS), PTEN-hamartoma Tumor Syndrome (PHTS) or Cowden Syndrome, and Juvenile Polyposis Syndrome (JPS), along with mixed polyposis and serrated polyposis [16–19].

In Tunisia, genetic investigation of hereditary CRC syndromes remains limited, and only a few studies have focused on LS or familial polyposis [20–23].

The aim of this study is to perform genetic profiling of patients selected according to their clinical features and family history suggesting hereditary polyposis or non-polyposis CRC syndromes. A targeted NGS cancer panel was used to identify germline pathogenic variants (PVs) to confirm the clinical diagnosis and improve the management of the patients and their family members.

## Patients and methods

### Patients

The study included 30 patients suspected of having hereditary polyposis or non-polyposis syndromes, such as MAP, FAP, and LS, according to the NCCN (National Comprehensive Cancer Network) guidelines. Briefly, patients with Polyposis syndromes, the criteria were: early age of onset of polyposis; diagnosis of numerous adenomatous polyps in the colon and rectum, and familial history of polyposis. For Lynch Syndrome patients, the criteria were: CRC and/or endometrial cancer diagnosed before age 50; other Lynch Syndrome-associated cancers such as ovarian, stomach, small bowel, hepatobiliary, upper urinary tract cancers, family history with at least 3 members with Lynch-associated cancers; diagnosis of cancer should be spread across at least 2 generations, and one should be diagnosed before age 50. For other familial Colorectal Cancer (without defined syndrome), we focused on family history with at least two first-degree relatives with CRC or advanced adenomas at <50 years of age.

Patients were recruited from the Department of Medical Genetics at Hedi Chaker Hospital and the Department of Surgery at Habib Bourguiba Hospital in Sfax, Tunisia, between June 2023 and May 2024.

### DNA extraction and targeted sequencing

Genomic DNA was isolated from peripheral blood leukocytes using the QIAamp DNA Blood Mini Kit, following the manufacturer's instructions (Qiagen). DNA was quantified using the Qubit 3.0 Fluorometric Quantitation (Thermo Fisher Scientific).

An aliquot of 100 ng of genomic DNA was used to prepare the library using the OncoRisk Cancer Panel, including 31 genes: *BRCA1, BRCA2, PALB2, BARD1, BRIP1, RAD51C, RAD51D, RAD50, NBN, MRE11A, ATM, CHEK2, TP53, PTEN, APC, BLM, BMPR1A, CDH1, CDK4, CDKN2A, EPCAM, MLH1, MSH2, MSH6, MUTYH, PMS2, PRSS1, SLX4, SMAD4, STK11,* and *VHL* (Celemics). The libraries were quantified using the Qubit® dsDNA HS Assay Kit (Life Technologies), then pooled and prepared for sequencing using the MiSeq Reagent Kit v3 (300 cycles) (Illumina, San Diego, CA). This procedure generated paired-end reads with a 151-bp read length. Reads were trimmed to remove low-quality sequences and then aligned to the human reference genome (GRCh37/hg19) using the Burrows-Wheeler Alignment (BWA) package. Sequences from the National Center for Biotechnology Information (NCBI) database (http://www.ncbi.nlm.nih.gov) were used as the reference.

### Analysis and classification of variants

Raw reads were analyzed using the cloud-based tool BaseSpace Variant Interpreter (https://basespace.illumina.com). During variant calling, modifications identified by the software were refined based on criteria including a coverage of

more than 100× and a variant allele frequency (VAF) of at least 5%. The genetic variants were classified into five classes: benign, likely benign, variant of uncertain significance (VUS), likely pathogenic, and pathogenic. Variants were referred according to the nomenclature recommendations of the Human Genome Variation Society (https://www.hgvs.org). Pathogenicity was assessed by comparing the data with sequence databases: Varsome (https://varsome. com), and ClinVar (https://www.ncbi.nlm.nih.gov/clinvar/).

## Sanger sequencing

Sanger sequencing was used to screen exon 13 of the *MUTYH* gene in MAP 'patients and exon 18 of the *APC* gene in FAP' patients as well as to confirm the presence of the variants identified by NGS. Forward and reverse primers were designed using Primer 3.0 software to amplify the fragments covering the variant regions and were provided upon request. PCR products were purified and labelled using the BigDye Terminator v3.1 Cycle Sequencing Kit and sequenced on the SeqStudio (Applied Biosystems). Sequence analysis was performed using BioEdit software.

## Multiplex ligation-dependent probe amplification (MLPA) assay

MLPA was performed to identify large deletions/duplications in the *STK11* gene. Briefly, 125 ng of genomic DNA was used as a template for MLPA with the SALSA MLPA Probemix P101 STK11 kit and SALSA MLPA Reagent Kit, according to the manufacturer's instructions (MRC Holland). Amplification products were then injected into a SeqStudio instrument (Applied Biosystems) along with the GeneScan™ 500 LIZ™ dye Size Standard to analyse the copy number variations in the corresponding gene. Results were interpreted using the Coffalyser.Net™ MLPA analysis software (MRC Holland).

## Results

### Patients

Our cohort includes 30 Tunisian patients with suspected hereditary polyposis (Group 1) or non polyposis syndromes (Group 2). Genetic testing was performed to confirm the diagnosis based on clinical features and family history, using Sanger sequencing or Targeted NGS (Fig 1).

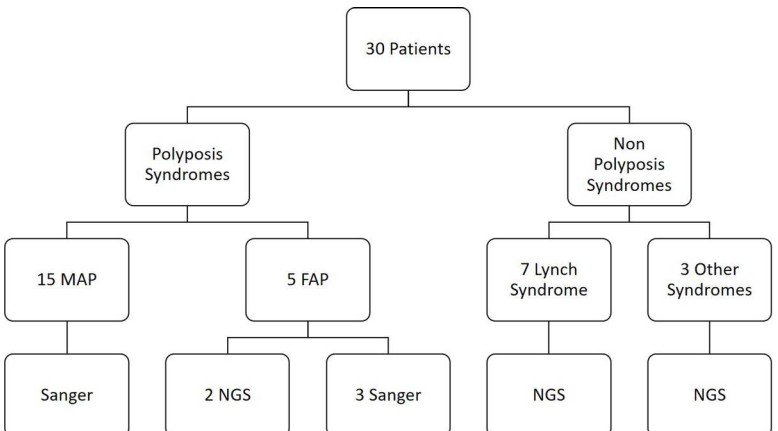

**Fig 1. Workflow showing group of patients according to their clinical features and family history of cancer, and the methodology used for genetic investigation.**

## Patients with hereditary polyposis syndromes

In this group, fifteen patients were suspected to have MUTYH-Associated Polyposis (MAP) and thus screened for pathogenic variants (PVs) in the *MUTYH* gene. In our previous work, we showed that the PV (c.1143_1144dup, p.Glu382fs) was recurrent in Tunisian MAP patients [20]. Therefore, we initially screened exon 13 of the *MUTYH* gene using Sanger sequencing. We found that almost all patients (12/15, 80%) were homozygous carriers for the c.1143_1144dup, p.Glu-382Ter variant, confirming the clinical diagnosis of MAP and the high prevalence of this PV in Tunisian MAP patients. Fig 2a showed an example of homozygous c.1143_1144dup variant in MP3 patient (Fig 2a, Table 1, S1a and b Fig).

Patient MP13 had a family history of colorectal cancer (CRC). He was diagnosed with polyposis and, at the age of 40, developed CRC. Sanger sequencing of exon 13 of *MUTYH* revealed heterozygous composite genotype (c.1143_1144dupGG/c.1103G>A). Furthermore, we identified combined genotypes in affected family members, such as

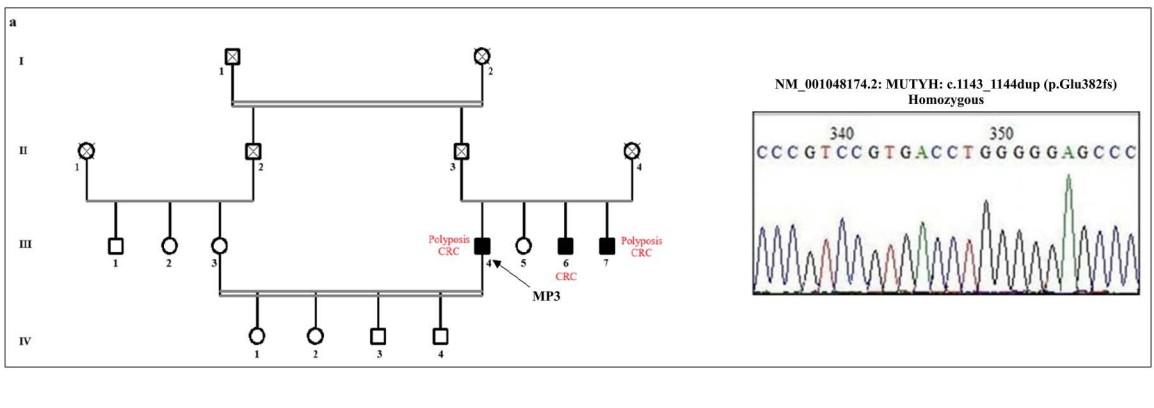

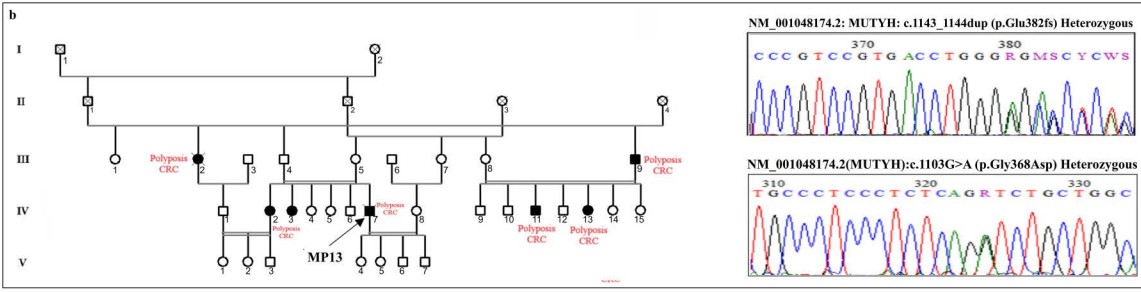

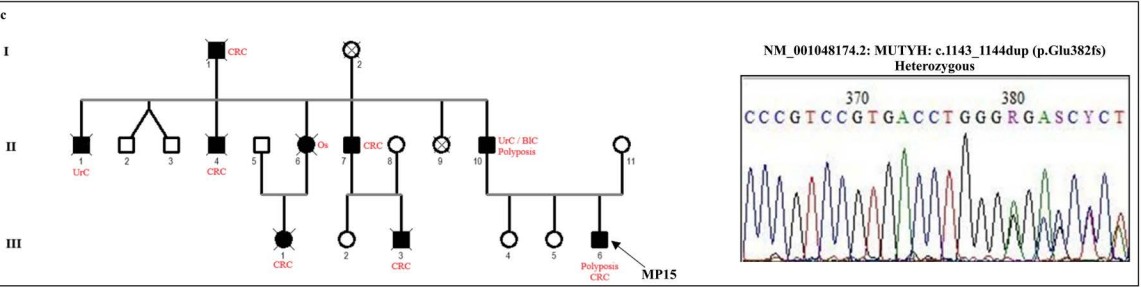

**Fig 2. Pedigrees of patients with MAP, and chromatograms showing the c.1143_1144dup PV in the *MUTYH* gene at homozygous (a), combined with the c.1103 G>A (b), and at heterozygous genotypes (c).** Males are indicated by squares, females by circles. An arrow indicates the proband, and cancer history in the family is indicated in red.

c.1143_1144dup/c.452A > G, and c.1103G > A/c.452A > G, while the non-affected individuals were all monoallelic carriers of either the c.452A > G or c.1143_1144dup variant (Fig 2b, Table 1). For MP14, and MP15 patients, Sanger sequencing of the *MUTYH* coding region showed that both were heterozygous for the c.1143_1144dup variant. We performed targeted-NGS and did not identify any pathogenic or likely pathogenic variants (P/LPVs) in other genes associated with hereditary polyposis syndromes (Fig 2c, Table 1).

In our cohort, five patients were clinically suspected to have Familial Adenomatous Polyposis (FAP). To confirm the diagnosis, we screened the *APC* exon 18 gene, and identified 2 PVs: FP1 and FP5 patients: the c.3183_3187del, p.Lys1061_Gln1062insTer, and c.2016–2017delTA, p.His672Ter respectively, confirming the FAP diagnosis. Fig 3a illustrates the c.3183_3187del, p.Lys1061_Gln1062insTer PV identified in FP1 patient. For other patients (FP2, FP3, and FP4), we conducted Targeted NGS analysis of germline DNA using the cancer panel. We found that FP3 and FP4 did not carry any P/LPVs in the 31 genes included in the

panel, however, FP2, carried a PV (c.502–503del, p.Ile168SerTer11) in *PTEN* gene (Fig 3b). This variant was reported twice in the ClinVar database and for the first time in a Tunisian patient. PVs in the PTEN gene are known to be associated with Cowden syndrome (CS). This patient presented with several serrated sessile polyps, which led pathologists to suspect FAP, but genetic testing accurately identified the syndrome, enabling better patient management and genetic counselling.

**Table 1. Clinical and genetic characteristics of patients with polyposis syndromes.**

**MAP**

| Patients | Age of onset of polyposis | N. of polyps | Family History | Gene | Variants |
|---|---|---|---|---|---|
| **MP1** | 34 | – | Polyp + CRC | *MUTYH* | c.1143_1144dup, p.Glu382Ter |
| **MP2** | 41 | – | // | | // |
| **MP3** | 44 | – | // | | // |
| **MP4** | 50 | 57 | // | | // |
| **MP5** | 47 | 6 | // | | // |
| **MP6** | 45 | | // | | // |
| **MP7** | 31 | <100 | // | | // |
| **MP8** | 47 | | // | | // |
| **MP9** | 41 | 20 | // | | // |
| **MP10** | 47 | 10 | // | | // |
| **MP11** | 20 | <100 | // | | // |
| **MP12** | 48 | <100 | // | | // |
| **MP13** | 44 | <100 | | | c.1143_1144dup, p.Glu382Ter c.1103G > A, p. Glu368Asp |
| **MP14** | 38 | – | UrC, Polyp+ CRC + Os + BlC | | c.1143_1144dup, p.Glu382Ter |
| **MP15** | 44 | – | Br, Bra | | // |

| | **FAP** | | | | |
|---|---|---|---|---|---|
| Patients | Age of onset of polyposis | N. of polyps | Extracolonic manifestation | Gene | Variants |
| FP1 | 25 | >100 | CHRPE + HC | *APC* | c.3183_3187del, p.Lys1061Gln1062insTer |
| FP2 | 56 | >100 | PpC + ThC + TC + KC | *PTEN* | c.502_503del, p.Ile168SerTer11 |
| FP3 | 65 | >100 | PrC | | – |
| FP4 | 62 | >100 | – | | – |
| FP5 | 26 | >100 | Polyp + CRC | *APC* | c.2016_2017del, p.His672Ter |

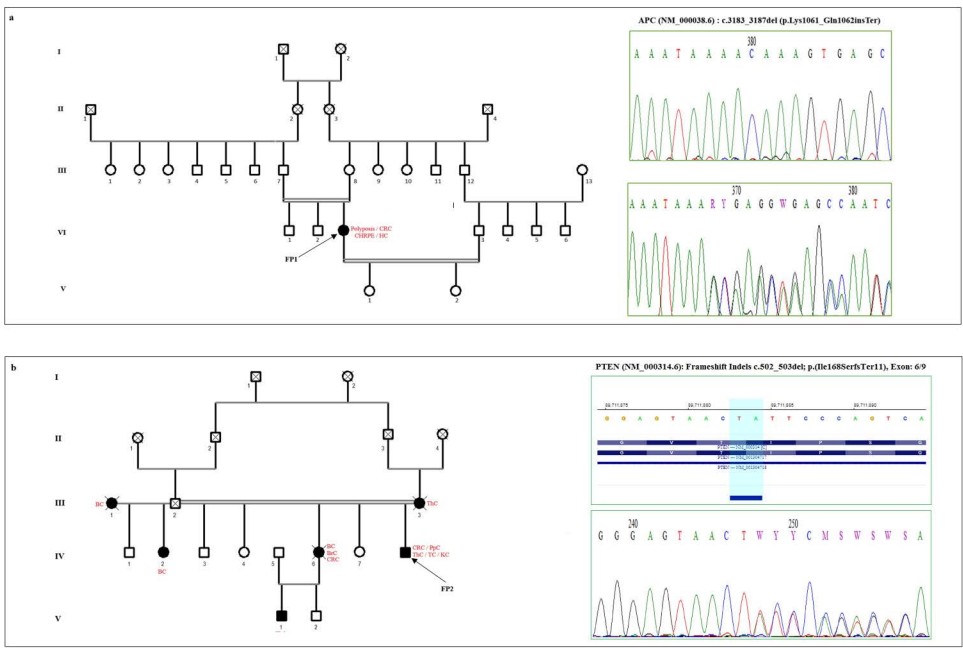

**Fig 3. Pedigrees of FP1 and FP2 patients showing PVs in (a) *APC* (c.3183_3187del), and (b) *PTEN* (c.502_503del) genes.** Probands are indicated by arrows; affected members are indicated in red.

## Patients with hereditary non-polyposis syndromes

Seven patients were suspected to have Lynch Syndrome (LS) according to the NCCN guidelines. NGS data revealed the absence of P/LPVs in the mismatch repair (MMR) genes associated with LS, and only variants of uncertain significance (VUS) were identified. We found that 2 patients (LSP2 and LSP7) carried a heterozygous missense variant (c.728G>A, p. Arg243Gln.) in the *MSH2* gene (Fig 4a, Table 2). This variant has been previously reported in LS patients and classified as conflicting of pathogenicity according to ACMG and ClinVar criteria. In patient LSP3, we identified a variant (c.1394G>A, p.Cys465Tyr) in the *BRIP1A* gene, which is not reported in gnomAD, and classified as a VUS according to ClinVar criteria (Fig 4c). The 3D protein models for both the mutated and wild-type forms of p.Arg243Gln and p.Cys465Tyr are shown in Fig 4b and d. Furthermore, VUS were also identified in other MMR genes, such as MLH1 (c.1217G>A, p.Ser406Asn) and BLM (c.3869C>T p. Ser1290Leu) in LSP1 and LSP6 patients, respectively (Table 2).

Interestingly, LSP4 was diagnosed with CRC at 34 years of age and had a strong family history of cancer (Fig 5a). We showed that LSP4 carried a variant (c.3500T>C) in exon 6 of the *MSH6* gene, which was confirmed by Sanger sequencing (Fig 5b). This variant was classified as VUS and is located in the P-loop domain containing the nucleoside triphosphate hydrolase. The c.3500T>C,p.Leu1167Pro variant has not been reported previously, although other variants at the same position (Leu1167Phe/His/Val) have been described. The introduction of a proline residue at this position destabilizes the MSH6 protein (ΔΔG = −1.677 kcal/mol), as predicted by the DynaMut2 tool (Fig 5c). Additionally, this variant was classified as damaging or probably damaging by other prediction tools, such as PolyPhen, SIFT, and FATHMM.

Immunohistochemical (IHC) analysis of MMR protein was performed for 6/7 available samples. Positive expression of the 4 MMR proteins was observed 4 patients (LSP1, LSP2, LSP3, and LSP7), whereas loss of MSH6 and MSH2/MSH6 was observed in LSP4 and LSP6 respectively (Table 2). Positive staining for MSH2, MLH1, PMS2, and a loss of MSH6 protein was illustrated in Fig 5d for LSP4 who carried the novel variant in *MSH6* gene (Fig 5d). Segregation analysis of this variant within the family and/or functional studies are needed to determine the pathogenicity of this novel VUS.

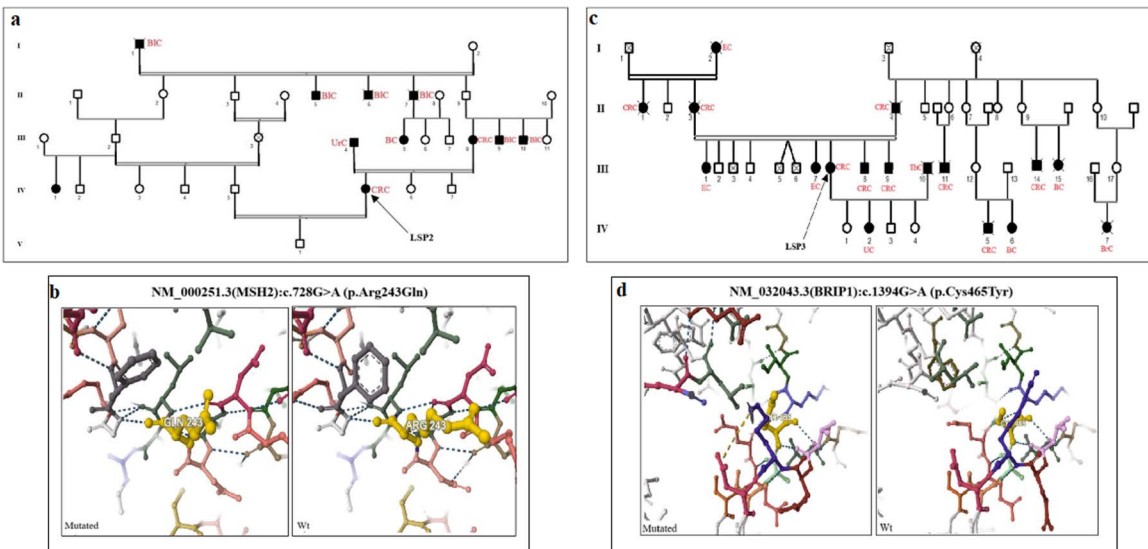

**Fig 4. Pedigrees of patients suspected of Lynch Syndrome: a (LSP2), and b (LSP3) carrying missense variants classified as Conflicting of Pathogenicity in *MSH2* (c.728G>A,p. Arg243Gln) and in *BRIP1A* (c.1394G>A,p.Cys465Tyr).** The 3D protein models for both the mutated and wild-type forms of **p.** Arg243Gln **(c)**, and **p.**Cys465Tyr **(d)** were presented.

**Table 2. Clinical and genetic characteristics of patients with non-polyposis syndromes.**

**Lynch Syndrome**

| Patients | Age of onset of CRC | Family History | MSI Testing | IHC MLH1/MSH2/ MSH6/PMS2 | Gene | Variants |
|---|---|---|---|---|---|---|
| LSP1 | 47 | GC, PC, BIC, BC | MSS | Presence | *MLH1* | c.1217G>A, p.Ser406Asn |
| LSP2 | 37 | UrC, BC, BlC, CRC | MSS | Presence | *MSH2* *SLX4* | c.728G>A, p. Arg243Gln c.832C>T, p.Arg278Trp |
| LSP3 | 62 | CRC, EC | MSS | Presence | *BRIP1A* | c.1394G>A, p.Cys465Tyr |
| LSP4 | 34 | LxC, BrC | MSS | Absence MSH6 | *MSH6* | c.3500T>C, p.Leu1167Pro |
| LSP5 | 36 | BrC+UC, L, LC | ND | ND | *ATM* | c.1516G>T, p.Gly506Cys |
| LSP6 | 44 | BC, PrC+CRC | MSI | Absence MSH2/MSH6 | *Check2* *BLM* | c.254C>T, p. Pro85Leu c.3869C>T, p. Ser1290Leu |
| LSP7 | 50 | CRC | MSS | Presence | *MSH2* | c.728G>A, p.Arg243Gln |

**Other Syndromes**

| Patient | Age of onset of CRC | Personal History | Family History | | Variant |
|---|---|---|---|---|---|
| OSP1 | 6 | | L, Os | *STK11* | deletion exons 2–10 |
| OSP2 | 12 | polyposis | CRC+OC UC,Br,C,L | *NBN* | c.425A>G,p.Asn142Ser |
| OSP3 | 33 | Polyposis | CRC | – | – |

## Genetic testing for rare syndromes (OSP)

Genetic testing using the cancer panel was performed on the germline DNA of three patients with rare syndromes (OSP), including Peutz-Jeghers Syndrome (PJS). Clinically, patient OSP1 was suspected of having PJS, which was confirmed by

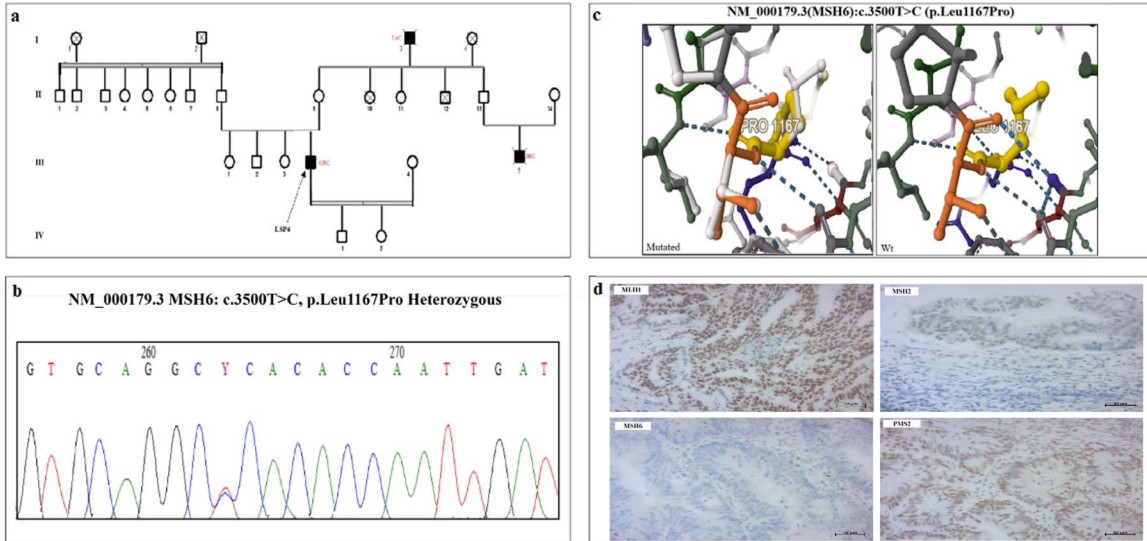

**Fig 5. (a) Pedigrees of patient LSP4 suspected of Lynch Syndrome, (b) chromatogram showing the c.3500T > C missense variant in MSH6 gene at heterozygous state, (c) The 3D protein models for both the mutated and wild-type forms of the p. Leu1167Pro in MSH6, (d) IHC showing the expression of MMR proteins: immunostaining was positive for MLH1, MSH2 PMS2, and negative for MSH6.** An arrow indicates the proband, and cancer history in the family is indicated in red.

genetic analysis. Since no P/LPVs in the *STK11* gene were detected, we performed MLPA and identified a large deletion encompassing exons 2–10 of the *STK11* gene (Fig 6). For the other two patients (OSP2 and OSP3), we did not identify any P/LPVs except a missense variant (c.425A > G, p.Asn142Ser) in exon 4 of the *NBN* gene in OSP2 (Table 2).

## Discussion

CRC is one of the most common cancers worldwide, and among risk factors, the genetic predisposition plays an important role. About one-third of CRC cases show familial clustering, but only 5–16% are associated with germline P/LPVs in CRC-predisposing genes [2–5]. Hereditary CRC syndromes are mainly classified into polyposis and non-polyposis types [7]. The most common polyposis syndromes are Familial Adenomatous Polyposis (FAP) and MUTYH-associated polyposis (MAP) [12–15]. There is limited information on the clinical and molecular characteristics of Tunisian patients with hereditary cancer syndromes, with only few studies reporting the spectrum of PVs, primarily in LS patients [20–23].

In this study, we used targeted NGS to analyse 30 Tunisian patients with polyposis or non-polyposis syndromes. Based on clinical features and family history, we selected 15 patients suspected to have MAP, 5 with FAP, 7 with LS, and 3 with other rare hereditary cancer syndromes. Overall, we identified the causal PV/LPV in 13 out of 15 MAP patients; among them, 12 (80%) carried the biallelic c.1143_1144dup variant, confirming that this variant is recurrent in Tunisian MAP patients, as previously reported [20]. Compound genotypes were identified in patient MP13 and 3 individuals from his family, in whom the c.1143_1144dup variant was combined with the c.1103G > A variant, which is a common Caucasian genotype [24]. Additionally, 2 patients (MP14 and MP15) were monoallelic carriers of the c.1143_1144dup variant. The risk of developing CRC in patients carrying a monoallelic PV in the *MUTYH* gene is still controversial. Some studies suggested that heterozygous carriers may be at an increased risk of CRC, while other did not found a significantly increased risk [25–27]. Barreiro et al. indicated that monoallelic *MUTYH* germline PVs can lead to tumorigenesis through a mechanism of loss of heterozygosity of the wild-type allele in the tumor, which could probably explain the case observed in our study [27].

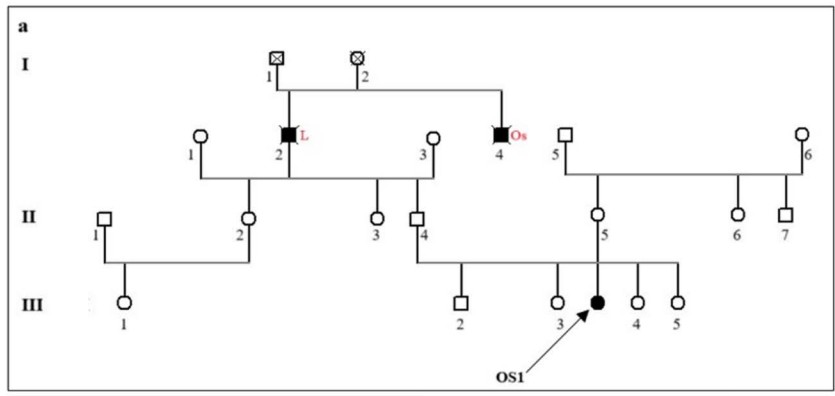

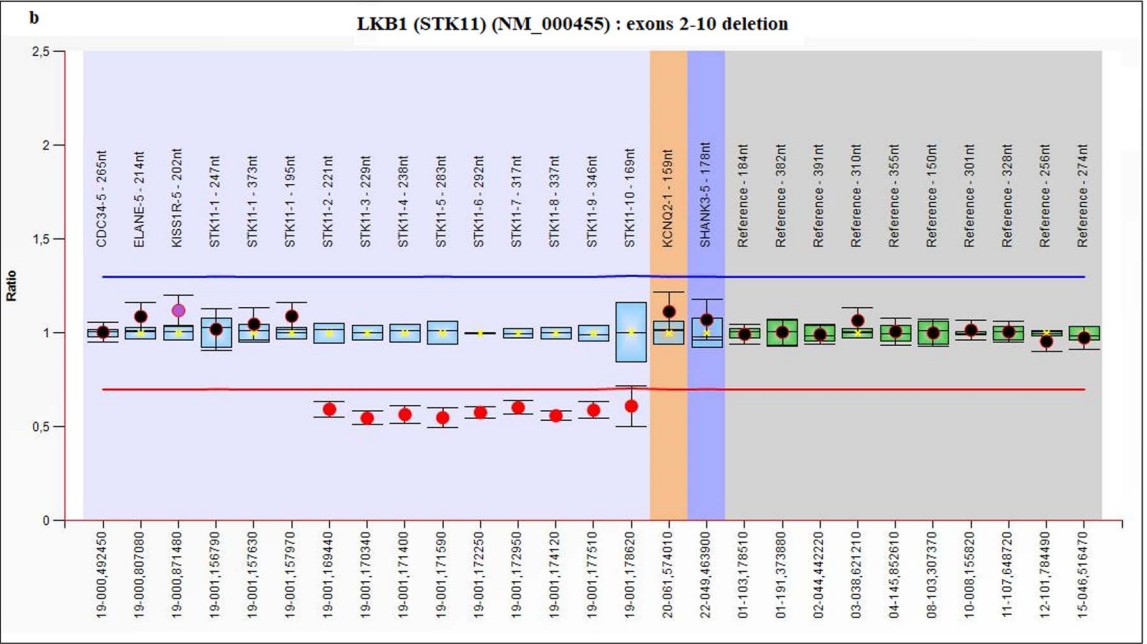

**Fig 6. (a) Pedigree of patient OS1 and (b) the MLPA result showing the CNV ratio chart in the STK11 gene.** An arrow indicates the proband, and familial cancer history is indicated in red.

In FAP patients, we identified the causal PVs (c.3183_3187del, p.Lys1061_Gln1062insTer, and c.2016_2017del, p.Ile168SerTer11) in exon 18 of the *APC* gene, in 2 out of 5 patients confirming thus FAP diagnosis. On the other hand, the clinical diagnosis for patient FP2, who was initially suspected of FAP, was corrected. Indeed, we identified a germline heterozygous PV in the *PTEN* gene (c.502–503del, p.Ile168SerTer11), which is a rare variant reported only twice in ClinVar and for the first time in a Tunisian patient. PTEN protein plays crucial roles in controlling the apoptosis and cell cycle by affecting the phosphatidylinositol 3-kinase (PI3K)/AKT/mammalian target of rapamycin (mTOR) pathways [28]. It is well documented that PVs in the *PTEN* gene are responsible for Cowden Syndrome/PTEN Hamartoma Tumor Syndrome (CS), a rare autosomal dominant syndrome that leads to the development of benign and malignant tumors involving the breast, thyroid, kidney, and uterus [29]. Early diagnosis of CS is challenging due to its variable expressivity and complexity. A recent study reported the case of a patient who developed the first sign of the disease (a peritonsillar polyp) at the age of 3, with the diagnosis of CS established later. By 18 years of age, hundreds of intestinal polyps were detected by

colonoscopy, and a *PTEN* germline frameshift variant c.762del, p.Val255Ter was identified, confirming the CS diagnosis [30]. Although currently, there is no specific treatment for CS, early diagnosis is essential for improving patient outcomes. However, a promising approach to restore the PTEN pathway through the use of mTOR inhibitors (Sirolimus) is currently under investigation [31].

In our cohort, 7 patients with clinical features indicating LS were selected for genetic analysis. NGS data showed that only VUS were identified in the MMR genes associated with LS. We identified a missense variant (c.728G>A, p.Arg-243Gln) in the *MSH2* gene in 2 unrelated patients. This variant has been previously reported in LS patients and classified as "variant of conflicting pathogenicity" according to the ACMG criteria and in the ClinVar database. The c.728G>A, p.Arg243Gln variant has been reported in young Tunisian and Algerian patients diagnosed with LS [21,32,33]. Recently, Kabbage et al. identified the *MSH2* (c.728G>A) variant in a Tunisian woman diagnosed with diffuse gastric carcinoma who fulfilled the international guidelines for LSII syndrome [34]. Based on structural prediction models and electrostatic potential calculations, the authors suggested that this variant may be classified as likely pathogenic, since it affects the MSH2-MLH1 complex as well as DNA complex stability [34]. Interestingly, three independent studies have reported the c.728G>A variant in the *MSH2* gene, suggesting that it might be a predisposing variant for the Tunisian population; however, further investigations are needed to confirm this hypothesis.

Furthermore, we identified a novel missense variant (c.3500T>C, p.Leu1167Pro) in the *MSH6* gene in one LS-suspected patient. This variant is classified as "conflicting pathogenicity" and localized in the P-loop domain of the MSH6 protein, which contains the nucleoside triphosphate hydrolase domain [35]. In silico analysis, showed that the proline residue at this position destabilizes the MSH6 protein, and several prediction tools indicate that this variant is damaging or probably damaging. IHC showed negative staining for MSH6, while MSH2, MLH1, and PMS2 proteins displayed positive expression, which is in line with the in silico analysis. However, further investigations, such as functional studies and familial segregation, are needed to confirm the pathogenicity of this variant. It is worth mentioning that the Leu1167Pro variant is described for the first time in this study, but other variants at the same position (p.Leu1167Phe/His/Val) have been reported in public databases.

Furthermore, we identified a VUS in the *BRIP1A* gene in one patient with a strong family history of cancer. Ali et al. suggested that germline *BRIP1A* variants may be associated with predisposition to CRC, and that patients carrying variants in *BRIP1A* gene may potentially had frequent colonoscopy screening, especially if there is a family history of CRC [36].

In the LS group, we didn't identify any P/LPVs in the MMR genes that are commonly associated with LS. It is well established that Familial Colorectal Cancer Type X (FCCTX) and LS share similarities in their clinical presentation, but they differ at the molecular level [37,38]. LS is caused by defects in mismatch repair (MMR) genes, whereas several genes are potentially associated with FCCTX, namely *BMPR1A, RPS20, SEMA4A, SETD6, BRCA2, OGG1*, and *FAN1* [39]. Only two genes (*BRIP1A* and *BRCA2*) were included in the panel used in this study, and further analysis with a larger cancer panel is necessary to investigate these patients.

Peutz-Jeghers syndrome (PJS) is an autosomal dominant inherited disorder characterized by intestinal hamartomatous polyps in association with a distinct pattern of skin and mucosal macular melanin deposition [40]. A young patient clinically suspected of having PJS was selected for genetic testing to identify the causative variant and confirm the clinical diagnosis. NGS data analysis showed no germline PVs in the *STK11* gene; however, a large deletion encompassing nine exons (from exon 2 to exon 10) was identified, confirming the PJS diagnosis in this patient.

Altogether, our results showed that among 30 young patients clinically diagnosed with hereditary CRC syndromes, we were able to identify PVs in 17 patients (13 cases with MAP, 2 cases with FAP, 1 case with CS, and 1 case with PJS). For other patients, the underlying genetic cause remains unknown, and further investigations, such as the use of a larger cancer panel or whole-exome sequencing, are needed for patient follow-up and genetic counselling for relatives.

## Conclusion

Our study contributes to a better understanding of hereditary CRC syndromes in Tunisian families, which may lead to improved patient and family management. Through this genetic analysis, we have either confirmed or corrected the clinical diagnosis for 17/30 patients. However, for others, we were unable to identify the causative germline PV despite clinical features indicating specific CRC syndrome and a strong family history of cancer. This suggests that further genetic investigations, such as the use of larger cancer panels or whole-exome sequencing, may be necessary for testing these patients.

## Supporting information

**S1 Fig. Pedigrees of other families suspected with the MAP (a, b), the FAP (c, d), LS (e, f, g, h), and the OS (i).** An arrow indicates the proband individual, and the familial cancer history is highlighted in red.
(JPG)

## Acknowledgments

We thank the patients for their cooperation and the technical support of N. Kchaou from CBS for Miseq and SeqStudio manipulation.

## Author contributions

**Conceptualization:** Rania Abdelmaksoud-Dammak.

**Data curation:** Rania Abdelmaksoud-Dammak, Nihel Ammous-Boukhris.

**Formal analysis:** Amena Saadallah-Kallel, Souhir Guidara, Imen Miladi-Abdennadher, Ikhlas Ben Ayed, Slim Charfi, Imen Zribi.

**Investigation:** Rania Abdelmaksoud-Dammak, Nihel Ammous-Boukhris, Raja Mokdad-Gargouri.

**Methodology:** Rania Abdelmaksoud-Dammak, Nihel Ammous-Boukhris, Amena Saadallah-Kallel, Dorra Ben Ayed-Guerfali, Souhir Guidara, Imen Miladi-Abdennadher, Ikhlas Ben Ayed, Slim Charfi.

**Resources:** Hassen Kamoun, Tahya Sellami-Boudawara, Imen Zribi, Foued Frikha.

**Supervision:** Salah Boujelbene, Raja Mokdad-Gargouri.

**Validation:** Rania Abdelmaksoud-Dammak.

**Writing – original draft:** Rania Abdelmaksoud-Dammak, Nihel Ammous-Boukhris, Amena Saadallah-Kallel, Raja Mokdad-Gargouri.

**Writing – review & editing:** Rania Abdelmaksoud-Dammak, Nihel Ammous-Boukhris, Amena Saadallah-Kallel, Souhir Guidara, Imen Miladi-Abdennadher, Ikhlas Ben Ayed, Hassen Kamoun, Slim Charfi, Tahya Sellami-Boudawara, Imen Zribi, Foued Frikha, Salah Boujelbene, Raja Mokdad-Gargouri.

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
