## [Decision Letter · Decision Letter 0]

Dear Dr. Mokdad-Gargouri,

Thank you for submitting your manuscript to PLOS ONE. After careful consideration, we feel that it has merit but does not fully meet PLOS ONE’s publication criteria as it currently stands. Therefore, we invite you to submit a revised version of the manuscript that addresses the points raised during the review process.

Please carefully read the comments from both reviewers and completely address their concerns.

We look forward to receiving your revised manuscript.

Kind regards,

Chunming Liu

Academic Editor

PLOS ONE

Journal Requirements:

No

Tunisian Ministry of Higher education and Scientific Research

6. We note that you have indicated that there are restrictions to data sharing for this study. For studies involving human research participant data or other sensitive data, we encourage authors to share de-identified or anonymized data. However, when data cannot be publicly shared for ethical reasons, we allow authors to make their data sets available upon request. For information on unacceptable data access restrictions, please see http://journals.plos.org/plosone/s/data-availability#loc-unacceptable-data-access-restrictions.

7. In the online submission form, you indicated that data are available from the corresponding authoy upon request.

8. Please amend your authorship list in your manuscript file to include author Dorra Ben Ayed Guerfali

9. Please amend the manuscript submission data (via Edit Submission) to include author Salah Boujelbene.

10. Please include your tables as part of your main manuscript and remove the individual files. Please note that supplementary tables (should remain/ be uploaded) as separate "supporting information" files.

Reviewers' comments:

Reviewer's Responses to Questions

**Comments to the Author**

1. Is the manuscript technically sound, and do the data support the conclusions?

Reviewer #1: Yes

Reviewer #2: Partly

2. Has the statistical analysis been performed appropriately and rigorously?

Reviewer #1: Yes

Reviewer #2: N/A

3. Have the authors made all data underlying the findings in their manuscript fully available?

Reviewer #1: No

Reviewer #2: Yes

4. Is the manuscript presented in an intelligible fashion and written in standard English?

Reviewer #1: Yes

Reviewer #2: Yes

Reviewer #1: In this manuscript the authors confirmed 17 of 30 Tunisian colorectal cancer patients by identifying pathogenic variants using Sanger sequencing or Targeted NGS with a 31-gene panel. Results highlighted the necessity of genetic testing in confirming the genetic causes of CRC syndromes, especially in Lynch Syndrome. Minor issues:

1, Page 7 in the Results section, “five teen patients” should be fifteen patients.

2, Inconsistency of labeling: in figure 2a, “MP3” should be “MP13”. In addition, In page 7, patients were referred as MAP13, MAP14, MAP15 but in page 10 and in figure 2 they were labelled as MP13, MP14, MP15.

3, In page 5, the extra dot should be removed in “. Multiplex Ligation-Dependent Probe Amplification”.

4, The NGS sequencing data such as raw reads and SAM files should be deposited to a public repository.

5, Since this study focused on CRC samples, it would be better if authors discussed why the OncoRisk Cancer Panel was used instead of targeted CRC panel which typically includes other classical CRC oncogenes such as BRAF, KRAS and CTNNB1.

Reviewer #2: This study assessed the impact of genetic testing in the diagnosis of 30 patients suspected with hereditary polyposis or non-polyposis syndromes.

1. The authors should clarify the criteria/guidelines used to consider patients as suspected of having hereditary polyposis or non-polyposis syndromes. For example, did the authors follow NCCN Guidelines for Genetic/Familial High-Risk Assessment?

2. It would be helpful to include additional data on the concordance between Sanger sequencing and targeted NGS. Specifically, as both Sanger sequencing and targeted NGS were performed for patients MAP14 and MAP15, did the targeted NGS detect the same MUTYH variant identified by Sanger sequencing?

3. I couldn’t find Table 1 and Table 2.

4. The authors should carefully proofread the manuscript to correct typos. Below are a few noted:

Abstract, “hereditary polyposis syndromes such as MUTY-Associated Polyposis” should be “… MUTYH…”

Page 6, “. Multiplex Ligation-Dependent Probe Amplification” should be “Multiplex Ligation-Dependent Probe Amplification”

Page 7, “In this group, five teen patients suspected of MYH-Associated Polyposis” should be “… fifteen…”

**Do you want your identity to be public for this peer review?** For information about this choice, including consent withdrawal, please see our Privacy Policy

Reviewer #1: No

Reviewer #2: No

---

## [Author Response · Author response to Decision Letter 1]

7 May 2025

We have addressed all comments made by the editor and reviewers, and updated our manuscript accordingly. The point by point response to reviewers was attached in files

---

## [Decision Letter · Decision Letter 1]

Genetic Profiling of Inherited Colorectal Cancer Syndromes in Tunisian Patients

PONE-D-25-11904R1

Dear Dr. Raja Mokdad-Gargouri,

We’re pleased to inform you that your manuscript has been judged scientifically suitable for publication and will be formally accepted for publication once it meets all outstanding technical requirements.

Kind regards,

Chunming Liu

Academic Editor

PLOS ONE

Additional Editor Comments (optional):

Reviewers' comments:

Reviewer's Responses to Questions

**Comments to the Author**

Reviewer #1: All comments have been addressed

Reviewer #2: All comments have been addressed

2. Is the manuscript technically sound, and do the data support the conclusions?

Reviewer #1: Yes

Reviewer #2: Yes

3. Has the statistical analysis been performed appropriately and rigorously?

Reviewer #1: Yes

Reviewer #2: N/A

4. Have the authors made all data underlying the findings in their manuscript fully available?

Reviewer #1: Yes

Reviewer #2: Yes

5. Is the manuscript presented in an intelligible fashion and written in standard English?

Reviewer #1: Yes

Reviewer #2: Yes

Reviewer #1: The authors addressed all my concerns.

Reviewer #2: The authors have addressed all my previous comments to a satisfactory level. I don’t have any further comments.

**Do you want your identity to be public for this peer review?** For information about this choice, including consent withdrawal, please see our Privacy Policy

Reviewer #1: No

Reviewer #2: No

---

## [Editor Report · Acceptance letter]

PONE-D-25-11904R1

PLOS ONE

Dear Dr. Mokdad-Gargouri,

I'm pleased to inform you that your manuscript has been deemed suitable for publication in PLOS ONE. Congratulations! Your manuscript is now being handed over to our production team.

Kind regards,

on behalf of

Dr. Chunming Liu

Academic Editor

PLOS ONE